# Microbial L-asparaginase for Application in Acrylamide Mitigation from Food: Current Research Status and Future Perspectives

**DOI:** 10.3390/microorganisms9081659

**Published:** 2021-08-03

**Authors:** Ruiying Jia, Xiao Wan, Xu Geng, Deming Xue, Zhenxing Xie, Chaoran Chen

**Affiliations:** 1Institute of Nursing and Health, College of Nursing and Health, Henan University, Kaifeng 475004, China; jry@henu.edu.cn (R.J.); wanxiao@henu.edu.cn (X.W.); 2School of Basic Medicine, Henan University, Jinming Avenue, Kaifeng 475004, China; 10190105@vip.henu.edu.cn; 3School of Life Science, Henan Normal University, Xinxiang 453007, China; 042061@htu.edu.cn

**Keywords:** acrylamide, L-asparaginase, microbes, immobilization, food processing, fermentation, purification

## Abstract

L-asparaginase (E.C.3.5.1.1) hydrolyzes L-asparagine to L-aspartic acid and ammonia, which has been widely applied in the pharmaceutical and food industries. Microbes have advantages for L-asparaginase production, and there are several commercially available forms of L-asparaginase, all of which are derived from microbes. Generally, L-asparaginase has an optimum pH range of 5.0–9.0 and an optimum temperature of between 30 and 60 °C. However, the optimum temperature of L-asparaginase from hyperthermophilic archaea is considerable higher (between 85 and 100 °C). The native properties of the enzymes can be enhanced by using immobilization techniques. The stability and recyclability of immobilized enzymes makes them more suitable for food applications. This current work describes the classification, catalytic mechanism, production, purification, and immobilization of microbial L-asparaginase, focusing on its application as an effective reducer of acrylamide in fried potato products, bakery products, and coffee. This highlights the prospects of cost-effective L-asparaginase, thermostable L-asparaginase, and immobilized L-asparaginase as good candidates for food application in the future.

## 1. Introduction

L-asparaginase (L-asparagine amidohydrolase, E.C.3.5.1.1), which was first mentioned by Clementi in 1922, catalyzes the hydrolysis of L-asparagine into aspartic acid and ammonia (Figure 1A) [1,2,3]. L-asparaginases have wide distribution among microbes, plants, and animals. Microbes are preferred over other sources of production of L-asparaginase because they have some advantages, such as easy upstream bioprocessing and convenient downstream processing, both of which facilitate industry-scale production [4]. Microbial L-asparaginase has been extensively studied in recent years for its potential applications in the pharmaceutical and food industries [4,5].

L-asparaginase is a prime therapeutic agent for the treatment of acute lymphoblastic leukemia (ALL) and lymphosarcoma. Certain cancers, such as acute lymphoblastic leukemia (ALL), have little or no L-asparagine synthetase enzyme [6]. As a result, L-asparagine in the blood becomes the sole source of this amino acid for ALL cells. Administration of L-asparaginase to ALL patients acts to deplete the blood of L-asparagine, which is non-essential for normal cells, depriving the ALL cells of this amino acid and ultimately leading to cell death. Elspar^®^, Oncaspar^®^, Erwinase^®^, and Kidrolase^®^ are commercially available forms of L-asparaginase medicinal drugs from bacteria (*E. coli* and *Erwinia chrysanthemi*) [7].

In 2002, researchers from Sweden discovered high contents of acrylamide in carbohydrate-rich foods subjected to elevated temperatures, highlighting the significant sources of dietary acrylamide intakes [8,9]. Acrylamide is formed between reducing sugars such as glucose and L-asparagine due to frying, baking, or grilling starchy foods at over 120 °C in low humidity conditions through a non-enzymatic process called the Maillard reaction [10,11,12,13]. The mechanism involves the formation of a Schiff base followed by the decarboxylation and elimination of either ammonia or a substituted imine under heat to yield acrylamide [14] (Figure 2A). Acrylamide has adverse effects on human health and has been proven to be neurotoxic, genotoxic, carcinogenic, and toxic to the reproductive system [15,16]. The International Agency for Research on Cancer [17] classified acrylamide as a probable human carcinogen (Group 2A), and the Scientific Committee on Food (SCF, 2002) reported that it was genotoxic [16]. Consequently, the European Commission [18] launched new reduction measures of acrylamide in food in 2017. Operators in the food business in Europe were obliged by Regulation 2158/2017 to reduce acrylamide content in their food products [19].

Acrylamide mitigation strategies were proposed, such as raw materials selection, product composition alteration, processing conditions optimization, and pretreatment procedures [19]. Enzymatic treatment, which is a simple and effective way to reduce acrylamide in food without affecting the sensory or nutritional properties of the final product, has been proposed [20]. L-asparaginase is a promising choice because of its hydrolytic activity toward L-asparagine into aspartic acid and ammonia in the treatment of foods. Consequently, aspartic acid cannot participate in the Maillard reaction, meaning acrylamide formation is significantly inhibited (Figure 2B) [21].

Acrylaway^®^ and PreventAse^®^ are commercially available forms of L-asparaginase from fungi (*Aspergillus oryzae* and *Aspergillus niger*) [5,22], which are safe and are recommended for use as food additives by the FAO/WHO Expert committee [22,23]. L-asparaginase has been granted “generally recognized as safe” (GRAS) status from the FDA [16]. Furthermore, L-asparaginase is deactivated during the heating process, ensuring its safe application in the food industry [24]. In this review, microbial sources of L-asparaginase are discussed: including classification, catalytic mechanism, production, purification and immobilization, focusing on its potential application for acrylamide mitigation in food such as fried potato products, bakery products, and coffee. The unique properties of immobilized L-asparaginase that make it more suitable for food application also are addressed.

## 2. Classification and Catalytic Mechanism of L-asparaginase

Based on their amino acid sequences and structural and functional homologies, L-asparaginases have been divided into three families: bacterial type (type I and type II) L-asparaginases, plant-type (type III) L-asparaginases, and Rhizobial-type L-asparaginases [25,26]. Bacterial type-I L-asparaginases are cytosolic enzymes with relatively low affinity towards L-asparagine (millimolar K_m_). Bacterial type-II L-asparaginases are periplasmic enzymes that exhibit high specific activity against L-asparagine (micromolar K_m_) [27]. Plant-type L-asparaginases are the members of the N-terminal nucleophile (Ntn) hydrolases superfamily. These enzymes are expressed as inactive precursors that undergo autocleavage to form α and β subunits. These subunits then form dimers of heterodimers to attain an active conformation. Plant-type L-asparaginases are subdivided into K^+^-dependent and K^+^-independent enzymes, depending on their potassium requirement for activation [28]. The third family includes sequences in homology to *Rhizobium etli* asparaginase II, of which *Rhizobium etli* (the symbiotic host of leguminous plants) can use L-asparagine as a sole carbon and nitrogen source through the action of two enzymes: asparaginase I and asparaginase II [29].

Despite being extensively studied, the mechanistic details of biochemical reactions catalyzed by L-asparaginase have been ambiguous. The hydrolysis of L-asparagine by L-asparaginases may have diverse mechanisms—the direct displacement or the double-displacement mechanism (Figure 1). The direct displacement mechanism is used by L-asparaginases, in which a water molecule directly attacks the L-asparagine substrate (Figure 1B). The double-displacement mechanism, commonly referred to as a ping-pong mechanism, involves an acyl type covalent intermediate, where the hydrolysis of L-asparaginases proceeds with two displacements. Each active site of L-asparaginases consists of two highly conserved catalytic triads (Thr-Tyr-Glu and Thr-Asp-Lys) represented by T16-Y29-E289 and T95-D96-K168, respectively, in *Helicobacter pylori* L-asparaginase [30,31]. At the beginning of the L-asparaginase reaction, the activated Thr of the first triad initiates a nucleophilic attack on the carbonyl amide in the L-asparagine side chain, resulting in the acyl-intermediate form of the enzyme and ammonia release (first displacement); then, the Thr of the second triad activates water molecules for a nucleophile attack on the same C atom, which breaks the acyl intermediate and releases aspartic acid (the second displacement) [30,32]. The direct displacement mechanism of catalysis by guinea pig L-asparaginase was supported by the experimental results and computational work [33]. The double-displacement (ping-pong) mechanism of catalysis by *E. coli* type II L-asparaginase was proven based on the structural and biochemical experiments combined with previously published data [34].

## 3. Sources of L-asparaginase

There are a variety of microbes, such as archaea, bacteria, actinomycetes, fungi, yeasts, and microalgae, that are sources of L-asparaginase. Microbes that produce L-asparaginase were isolated from the abundant environment such as soil, water, marine sediment, river sediment, marine sponges, mosses, and plants. At the same time, the variety of recombinant L-asparaginases was also produced by the L-asparaginase gene cloned and expressed in *E. coli* was based on protein engineering and recombinant DNA technology. Bacterial and fungal strains that are important sources of this enzyme have been extensively studied. Some of the recently used L-asparaginase producers, such as archaeal, bacterial, actinobacterial, fungal, yeast, and algal strains, are summarized in Table 1.

L-asparaginase-producing microbial strains can be investigated by using an agar plate assay and a thin layer chromatography (TLC) assay. El-Naggar and El-Shweihy (2020) evaluated L-asparaginase production from actinomycetes by using the agar plates with culture medium at a pH of 6.8–7, which consisted of L-asparagine and phenol red as indicators of pH value. L-asparaginase activity resulted in hydrolyzing L-asparagine into aspartic acid and ammonia that converted the phenol red from yellow to pink (alkaline condition) [35]. However, this method may produce false positives results [36]. Dias et al. (2019) used a TLC assay to screen fungal L-asparaginase producers. Fungal L-asparaginase activity was indicated by the formation of spots corresponding to the retention factor of L-aspartic acid in the TLC assay [37].

The identification of L-asparaginase-producing microbial strains is usually performed using 16S rRNA gene sequencing, internal transcribed spacer (ITS) sequencing, and 28S rRNA gene sequencing. The L-asparaginase-producing bacteria were identified as *Stenotrophomonas maltophilia* EMCC2297 using 16S rRNA sequencing [38]. The L-asparaginase-producing fungus was identified as *Penicillium crustosum* NMKA 511 using nuclear ribosomal DNA ITS sequencing [19]. The L-asparaginase-producing yeast was identified as *Leucosporidium muscorum* using 28S rRNA gene sequencing [39].

## 4. Production of L-asparaginase

The industrial production of L-asparaginase by different microbes was previously achieved through submerged fermentation (SF) [77]. Although SF is a well-established technology applied in large-scale industrial enzyme production due to better monitoring and ease of handling, it has several disadvantages, such as the generation of large amounts of wastewater, difficulties in effluent treatment processes, and a high production cost [78,79]. The performance of SF can be influenced by various factors such as the carbon and nitrogen sources, temperature, pH, fermentation time, agitation, and inoculum size [80,81]. Soliman et al. (2020) studied the impact of sixteen different independent factors, including carbon sources, energy sources, metals, nitrogen sources, agitation speed, temperature, inoculum size, incubation time, and pH, on the production of L-asparaginase by *Streptomyces fradiae* NEAE-82 by applying a Plackett–Burman statistical design, and found that L-asparagine was the most significant positive independent factor (*p*-value 0.0092) affecting L-asparaginase production [55]. L-asparagine is an inducer in the production of L-asparaginase [82,83]. In addition, the expression of recombinant L-asparaginases under control of the lac operator can be induced by isopropyl-β-D-thiogalactoside (IPTG) [25].

The production of L-asparaginase through different microbes can also be carried out by solid-state fermentation (SSF) [81,84]. SSF is an attractive alternative for SF for the production of enzymes and has several advantages, such as the relatively easy recovery of end products, a lower need of water, and fewer energy requirements [85,86]. In addition, SSF utilizes low-cost agro-industrial residues as the substrate, which act as both a source of nutrients and physical support for enzyme production [87]. Major factors affecting SSF success include the selection of a suitable microbe and substrate, moisture, pH, temperature, inoculum size, particle size, and incubation periods. Meghavarnam and Janakiraman (2017) studied the effect of various factors such as solid substrates (sixty-five kinds of agro-based materials), incubation periods, moisture, particle size, inoculum size, pH, and temperature on L-asparaginase production by *Fusarium culmorum* (ASP-87) under SSF. A maximum yield of L-asparaginase was obtained (18.91 U/gds) under optimized conditions [77]. SSF can also be carried out on inert support materials, such as polyurethane (PUF), and the fermentation scale of SSF can be amplified up to a kilogram level, even in the laboratory [70,88].

Each microbe or strain has its own special conditions for maximum enzyme production; therefore, the optimization of components of the fermentation medium and culture parameters is essential in upstream bioprocessing [80,89]. The one factor at a time (OFAT) approach can only affect a single factor at once and keep the remaining factors constant, but it neglects the effects of interactions among factors [77,81]. El-Gendy et al. (2021) applied the one factor at a time approach to optimize the production of L-asparaginase by *Fusarium equiseti* AHMF4 under submerged fermentation. The optimized fermentation conditions resulted in the maximum L-asparagine activity of 40.78 Um/L [65]. Response surface methodology is a collection of mathematical and statistical techniques based on the fit of a polynomial equation to the experimental data, which must describe the behavior of a data set with the objective of making statistical previsions [90]. Soliman et al. (2020) applied a Box–Behnken design to optimize the three significant factors: L-asparagine, pH, and NaCl. The optimized fermentation conditions resulted in a maximum L-asparagine activity of 53.572 Um/L. The authors observed that the production of L-asparaginase increased 3.41-fold over the statistical optimization study [55].

## 5. Purification and Biochemical Properties of Microbial L-asparaginase

The purification of L-asparaginase is a key procedure in its production and the basis of its application. Most purification procedures are performed by using conventional methods, such as ammonium sulphate precipitation combined with molecular exclusion chromatography or ion/anion exchange column chromatography [3]. Intracellular L-asparaginase purification requires more processes than extracellular L-asparaginase [4]. The broth culture after fermentation is separated into supernatant and cell pellet by centrifugation. The supernatant is a crude enzyme preparation of extracellular L-asparaginase. The cell pellet is lysed by sonication under cooling conditions, then disrupted cellular debris and unbroken cells are removed by centrifugation and the supernatant is collected for crude preparation of the intracellular enzyme [38,48]. The aqueous two-phase system (ATPS) is a liquid–liquid fractionation method used as a pre-purification method. Cardoso et al. (2020) purified L-asparaginase production of the fungus *Penicillium* sp.–encoded 2DSST1 through ATPMS using Triton X-114, and obtained a purification factor of 1.4 and a yield of 100% [62]. Nickel affinity chromatography is also used to purify the recombinant L-asparaginase, with a polyhistidine tag on either terminus [20]. Various methods of the purification of microbial L-asparaginase are listed in Table 2.

L-asparaginases from different microbes vary in their biochemical properties. Generally, the optimum temperature for L-asparaginase activity is between 30 and 60 °C; however, it is remarkable that the optimum temperature for L-asparaginase activity from hyperthermophilic archaea is between 85 and 100 °C (Table 2). The optimum pH for L-asparaginase activity is between 5.0 and 9.0 pH (Table 2). However, L-asparaginases from *Halomonas elongate* IBRC-M10216 and *Aspergillus oryzae* CCT 3940 show optimum activity in a wide pH range from 6 to 9 and 5 to 8, respectively (Table 2) [37,54].

Kinetic parameters are vital to be understood in order to use L-asparaginase efficiently in industrial processes [91]. The Michaelis constant (K_m_) is equivalent to the concentration of the substrate at which the reaction takes place at one half its maximum rate, which gives an approximation of the relative affinity between the enzyme and its substrate [92]. V_max_ is defined as the maximum rate at which an enzyme is fully saturated with its substrate concentration [93]. Turnover number (k_cat_) is defined as the maximum number of chemical conversions of substrate molecules per second that a single active site will execute for a given enzyme concentration for enzymes with two or more active sites. The K_m_ values of L-asparaginases from different microbes vary mostly from 0.012 mM to 35 mM (Table 2). Kinetic properties depend upon factors such as pH, temperature, type, and the concentration of substrate [5]. The details of these kinetic parameters for L-asparaginase from different microbes are listed in Table 2.

L-asparaginase from different sources exists in monomeric, dimeric, tetramer, and hexameric forms. The variation in molecular weights of L-asparaginases from different microbes is from 32 to 205 kDa, as seen in Table 2. Sindhu and Manonmani (2018) demonstrated that recombinant L-asparaginase was a homotetramer, with the molecular weight of the monomer being 35 kDa by SDS-PAGE, and a whole protein molecular weight of 140 kDa from gel filtration (TSKgel G3000 SWXL column) [94].

## 6. Application of Microbial L-asparaginase in Food

According to the European Food Safety Authority (EFSA), foods related to human-consumed acrylamide are primarily fried potato products, bakery products, and coffee, and the intake of acrylamide in diets is estimated to be between 0.3 and 1.9 μg/kg body weight [81]. Zyzak et al. (2003) first reported the application of L-asparaginase for acrylamide reduction in a potato matrix [14]. Currently, acrylamide mitigation in various foods, such as fried potato products, bakery products, and coffee, through microbial L-asparaginase pretreatment has become a promising research field in the food industry.

### 6.1. Fried Potato Products

Fried potato (*Solanum tuberosum*) products are widely consumed by millions of people throughout the world, and they have contributed to 50% of humanity’s ingestion of acrylamide in European countries [98,99]. The potential for acrylamide formation in potatoes is related to several factors, such as the L-asparagine content, reducing sugars, potato variety, frying oil, and the use of additives [99,100,101]. In potatoes, the concentration of L-asparagine is excessive compared with reducing sugars [102]. Therefore, the high L-asparagine content in potatoes is most likely responsible for acrylamide formation in fried potato products [100,103].

L-asparaginase can effectively inhibit acrylamide formation in fried potato products because of its catalytic hydrolysis of L-asparagine to aspartic acid. Jiao et al. (2020) used recombinant type-I actinobacterial L-asparaginase (AsAsnase) expressed by *E. coli* BL21 (DE3) to reduce acrylamide in potato chips. Fresh potato slices (2 mm) were washed with deionized water, soaked in different concentration of AsAsnase solution at 37 °C for 30 min, dried out on absorbent papers, and fried at 170 °C for 5 min. L-asparagine and acrylamide contents of the potato sample treated with 30.0 IU/mL of enzyme reduced by 22.0% and approximately 55.9%, respectively, compared to the control. The authors believed that the poor thermostability of AsAsnase (half-life at 40 °C of 9.63 min) might be responsible for its unsatisfactory acrylamide degradation rate [45].

Thermostable L-asparaginase has potential application in fried potato products. Zuo et al. (2015) used a recombinant thermostable archaeal L-asparaginase from *Thermococcus zilligii* AN1 TziAN1_1 expressed by *E. coli* BL21 (DE3) to reduce acrylamide in French fries. The thermostable L-asparaginase displayed maximum activity at pH 8.5 and 90 °C, and retained 70% of its original activity after 2 h of incubation at 85 °C. When potato samples were treated with 10 U/mL of L-asparaginase at 80 °C for only 4 min, which was then followed by frying at 175 °C for ~5 min, acrylamide content in the French fries was reduced by 80.5% compared with the control. The authors found that the acrylamide content in French fries decreased upon increasing the L-asparaginase dosage and treatment time [12]. The application of the non-thermostable L-asparaginases to reduce acrylamide formation in food was performed at no more than 60 °C. However, thermostable L-asparaginase treatment at high temperature for a short period is considerably more time-efficient for the reduction in acrylamide content.

Using L-asparaginase to mitigate acrylamide in fried potato products was tested under different conditions. Bhagat et al. (2016) used L-asparaginase from the endophytic bacteria *Pseudomonas oryzihabitans* to reduce acrylamide in potato slices. Three different procedures were used to treat potato slices with the same enzyme concentration (2.8 U/g dried potato): immersion in enzyme solution at room temperature for 30 min, blanching in enzyme solution at 60 °C for 15 min in combination with immersion in distilled water at room temperature for 60 min, and blanching in enzyme solution at 40 °C for 15 min. The dried potato slices were fried at 175 °C for 10 min. Compared with untreated controls, the reductions in acrylamide content in potato slices treated with the three different procedures were 54%, 28%, and 90%, respectively [53]. The temperature of L-asparaginase treatment was a key factor in acrylamide mitigation. The temperature of the enzyme treatment should be chosen to align with the properties of the enzyme.

Different types of L-asparaginase can be applied in fried potato products to prevent acrylamide formation. Shi et al. (2017) applied recombinant bacterial Rhizobial-type L-asparaginase from *Paenibaeillus barengoltzii* CAU904 (PbAsnase) expressed by *E. coli* BL21 (DE3) to reduce acrylamide in potato chips. Compared with the control, the acrylamide content in potato chips decreased by 86% after treatment by using L-asparaginase (80 U/mL/45 °C/20 min) in combination with conventional blanching (85 °C/3.5 min), facilitating the diffusion of the enzyme into potato tissues [104]. In another study, Sun et al. (2016) used recombinant bacterial type-II L-asparaginase from *Aquabacterium* sp. A7-Y expressed by *E. coli* BL21 (DE3) to reduce acrylamide formation in potato strips. Compared with untreated potato strips, 88.2% of acrylamide was removed in the 32 μg/mL enzyme-treated group [20]. There may be no significant difference between Rhizobial-types of L-asparaginase and type-II L-asparaginase in preventing acrylamide formation in fried potato products.

Moreover, L-asparaginase can penetrate through the cell wall weakened by ultrasonic energy according to the patent (RU2423876C2) which involved its application in fried potato products to reduce acrylamide [105]. Potato slices (about 0.053 inches thick) were treated with 100,000 units of L-asparaginase and ultrasonic energy at a frequency of (68 kHz) in an ultrasonic soaking machine at 78 °F for 40 min, and the L-asparagine concentration were reduced by 22% compared to potato slices only treated with the enzyme. The energy of ultrasonic radiation was applied for the potato slices from 30 s to about 60 min.

### 6.2. Bakery Products

Bakery products are staple foods in many countries, because they supply a considerable portion of nutrients such as proteins, carbohydrates, fiber, and vitamins [106]. In Europe, 20% of the intake of acrylamide has been attributed to bakery products because they contain acrylamide, which is formed through the Maillard reaction [98,107,108]. Many factors, such as the concentration of the precursors, flour quality, fermentation conditions, thermal processing methods, processing conditions, physical state of food, and additives, can affect the acrylamide content in bakery foods [98,109,110]. L-asparagine is a limiting factor for the formation of acrylamide in bakery products [107].

L-asparaginase was efficient in inhibiting the formation of acrylamide as well as decreasing L-asparagine. Meghavarnam and Janakiraman (2018) used fungal L-asparaginase from *Fusarium culmorum* (ASP-87) to reduce acrylamide in sweet bread. The dough treatment with L-asparaginase was fermented at 30 °C for 90 min, then remixed and fermented again for 25 min, molded in bread making trays, and baked at 220 °C for 25 min. Acrylamide content in baked bread remarkably decreased with the increase in enzyme concentration from 50 to 300 U/mL. The L-asparagine and acrylamide contents in sweet bread samples treated with 300 U/mL enzyme were reduced to 78% and 86%, respectively, compared to the control [108]. The level of L-asparagine in sweet bread directly corresponded to the acrylamide content.

The thermostable L-asparaginase also had potential application in bakery products. Hong et al. (2014) used recombinant archaeal L-asparaginase (Tk1656) from *Thermococcus kodakarensis* KOD1 (TkAsn) expressed by *E. coli* BL21 (DE3) to reduce acrylamide in baked dough. TkAsn displayed high thermal stability at 90 °C, with an insignificant loss in enzymatic activity. Dough treated with 1000 U/g of TkAsn was incubated at 60 °C overnight and then baked at 180 °C for 30 min. The acrylamide content in baked dough by treatment with TkAsn was declined to 60% compared wiht the control [42]. TkAsn has high thermal stability, but a reason for the smaller reduction in acrylamide content in dough might be because the application of L-asparaginase in dough preparation lacks some processes, such as remixing dough, re-fermentation, and dough molding, which is not conducive to the contact between the enzyme and the substrate.

There are some factors such as cooking temperature, time, and L-asparaginase concentration, which have a huge impact on reducing acrylamide content in bakery products. Matouri and Alemzadeh (2018) used L-asparaginase (Elspar^®^) to reduce acrylamide in yeast-leavened wheat bread. The authors found that L-asparaginase had an important role in reducing the effect on acrylamide formation, whereas baking temperature significantly increased the acrylamide content in bread. Three parameters of the cooking temperature and time, as well as enzyme concentration, have been optimized using a response surface methodology. An L-asparaginase level of 752.15 U/Kg, a baking temperature of 245.71°C, and a baking time of 14.55 min was the optimal combination for obtaining the minimum acrylamide concentration in bread [109]. Although L-asparaginase (Elspar^®^) is used to treat ALL, it still has potential applications for reducing acrylamide formation in bread.

Furthermore, the method involving the application of L-asparaginase in bakery products to reduce acrylamide comprised the sequential steps according to the patent (US8124396B2) [110]: providing the dough with a water content of 10–90% by weight at pH of 5–7, treating the dough with L-asparaginase, and heat-treating it to reach a final water content below 35% by weight. L-asparaginase may be added in an amount of 10–100 units per kilogram of dry matter, which is effective to reduce the amount of acrylamide in the final product.

### 6.3. Coffee

The most consumed beverage is coffee; over two billion cups of coffee are consumed worldwide on a daily basis [111,112,113]. The two main coffee species used in the preparation of the beverage are arabica (*Coffea arabica*) and robusta (*Coffea canephora var. robusta*) [81]. The ingredients of coffee include some bioactive compounds such as caffeine, caffeic acid, and chlorogenic acid [114,115]. Nevertheless, the roasting of coffee beans can form the toxic and carcinogenic compound acrylamide [99,116]. The high average acrylamide levels in coffee substitutes, roasted coffee, and instant coffee were 890, 256, and 229 µg/kg, respectively [117,118]. Coffee significantly contributes to the total acrylamide content in the diet [16,119]. The roasting process is the exclusive cause of high acrylamide content in roasted coffee beans [120].

Using the asparaginase enzyme is a promising strategy for reducing acrylamide content in roasted coffee beans. Khalil et al. (2021) used fungal L-asparaginase from *Penicillium crustosum* NMKA 511 (PcAsnase) to reduce acrylamide in roasted coffee beans. Green Arabica coffee beans (5 g) were steamed at 100 °C for 45 min and then incubated in PcAsnase (2 U/g of beans) solution at 35 °C at 20 rpm for 1 h, which enabled good access of L-asparaginase to the free L-asparagine inside the beans instead of just on the surface. Light coffee beans were obtained by roasting at 210 °C for 20 min, whereas dark coffee beans were obtained by roasting at 240 °C for 25 min. Compared with the control, the acrylamide content of light roasted beans and dark roasted beans treated with the enzyme were reduced to 80.7% and 75.8%, respectively. Furthermore, the amount of L-asparaginase used to reduce acrylamide in roasted coffee beans was considered safe based on the result of a cytotoxicity assay [19].

L-asparagine, the precursor for acrylamide formation, is a limiting factor for acrylamide content in coffee. Akgün et al. (2021) applied L-asparaginase (Acrylaway^®^) to reduce acrylamide in roasted green coffee beans (*Coffea arabica*). The optimized conditions for enzyme treatment in green coffee beans and the impacts of two independent variables (enzyme dosage and treatment time) on five dependent variables, including acrylamide, were investigated by employing a response surface methodology. The authors found that L-asparagine level is more determinant of acrylamide formation in coffee than sugar levels, and enzyme concentration more significantly impacted the acrylamide concentration of coffee than did the time of treatment [118].

It is vital to maintain the concentrations of beneficial compounds in coffee without compromising the sensory quality and health effects of the final product during treatment with L-asparaginase. Corrêa et al. (2021) used L-asparaginase (Acrylaway^®^) to reduce acrylamide in roasted coffee beans (*Coffea arabica*). The coffee beans were pretreated with flowing steam (100 °C) for 30 min, subjected to enzyme treatment under the optimum conditions (water content of 30%, enzyme load of 5000 ASNU/kg, at 60 °C at 200 rpm for 2 h), and then roasted at 230 °C for 15 min. The acrylamide content in the enzyme-treated coffee was reduced to 59% compared with the control sample (with no type of treatment) and 77% compared to the blank sample (without the addition of asparaginase). It was important that the L-asparaginase treatment did not influence the major bioactive compounds in coffee, such as caffeine, chlorogenic acid, and caffeic acid [16].

In addition, the method of applying L-asparaginase in the roasted coffee beans to reduce acrylamide comprised a series of processes according to the patent (JP4467517B2) [121]: pretreating the coffee beans, adding L-asparaginase to the coffee beans, providing sufficient time for the enzyme to react with L-asparagine, deactivating or removing the enzyme; and roasting the coffee beans. The pretreatments included rinsing, pressurization, steaming, blanching, heating, vacuum processing, particle size reduction, or a combination of these. The amount of enzyme added may depend on the degree of acrylamide reduction and the properties of the enzyme. The methods of adding the enzyme to the coffee beans include spraying, dipping, sprinkling, and a dominant bath. The amount of time for the enzyme to react with L-asparagine depends on the specific characteristics of coffee beans, the desired degree of acrylamide reduction, and the amount of specific enzyme. L-asparaginase can be deactivated through heating, pH adjustment, and protease treatment.

### 6.4. Industrial Processes

Many applications of L-asparaginase in food have been described on a laboratory scale to reduce the concentration of acrylamide, but simple transfer to an industrial scale is usually problematic [122]. The application of L-asparaginase in industrial processes to achieve a significant reduction in acrylamide in the final product (such as French fries) is of commercial significance. Operation steps, enzyme characteristics, time, temperature, and pH are the crucial parameters for the successful application of L-asparaginase in industrial processes.

Medeiros Vinci et al. (2011) evaluated the application of L-asparaginase (Acrylaway^®^) in chilled French fries (not par-fried) to reduce the concentration of acrylamide on an industrial production line with a capacity of 4 tons/hour. Chilled French fries treated with different enzyme concentrations in a dip tank were produced in 100 kg batches, then packed in 2.5 kg bags under a modified atmosphere (with increased concentrations of CO_2_) and stored at 4 °C. There is no par-frying process step that could deactivate the enzyme, therefore, L-asparaginase could remain active in the production of chilled French fries during the shelf life of the product. The L-asparagine of chilled French fries, treated with an enzyme concentration of 2500 ASNU/L, was completely converted after 4 days. When chilled French fries were par-fried at 140 °C for 3.5 min, cooled to room temperature, and then fried for 3 min at 160 °C after 4 days of storage, the acrylamide concentration of the enzyme-treated French fries was below the limit of detection (12.5 μg/kg), and the sensory properties of the final product were no affected [123].

Chilled French fries (not par-fried) represent a specific segment of the market that is different and narrower compared to frozen par-fried French fries. Rottmann et al. (2021) evaluated applications of L-asparaginase in par-fried French fries to alleviate acrylamide according to industrial conditions (a batch of 15 tons of potatoes). Potato strips (cut to sizes of 8 × 8 mm) in the floatation channel were treated for 1 min in different L-asparaginase (PreventASe L^®^) concentration solutions combined with 0.045 mmol/L sodium dihydrogen diphosphate (SAPP) at pH 5 and 60 °C. Afterward, potato stripes were dried (80 °C, hot air), par-fried, quick frozen, and packed. All processing steps (pre-treatments) were carried out in a production plant and final frying proceeded at 175 °C for 3 min. The acrylamide concentration of French fries treated with enzyme concentrations of 1%, 0.5%, 0.3%, and 0.1% were reduced by 59%, 57%, 45%, and 27%, respectively, and the color or taste of the final product was not affected. The authors found that the hydrolytic activity of L-asparaginase under actual conditions was only one-third (low enzyme concentration) to less than one quarter (high enzyme concentration) of its theoretical activity. The authors believed that the use of a 0.5% enzyme solution resulted in a reasonable cost/effect ratio. Depending on the enzyme activities used, the additional costs of 0.05 (for 0.1% enzyme) to 0.5 USD/kg (for 1.0% enzyme) of French fries were estimated [122].

## 7. Application of Immobilized L-asparaginase in Food

Most studies of immobilized L-asparaginase have focused on its medical applications, while only a few works have been devoted to the food industry [124]. The stability and reusability of L-asparaginase are both essential from an industrial point of view because of the enzymes subjected to incubation and pretreatment at elevated temperatures. The poor stability and non-recyclability of free L-asparaginase increase the cost of food processing and limit the large-scale usage of the enzyme. Many materials, such as chitosan, aluminum oxide pellets, magnetic nanoparticles, and agarose spheres, have been utilized to immobilize L-asparaginase in order to increase the stability and recyclability of L-asparaginase.

The native properties of the enzymes, such as thermal properties, activity, and substrate affinity, can be enhanced for food applications using immobilization techniques. Moreover, the unique reusability of an immobilized enzyme makes it more suitable for food applications. Alam et al. (2018) used magnetic nanoparticle-immobilized L-asparaginase to reduce acrylamide in a starch–asparagine food model. L-asparaginase from *Bacillus aryabhattai* was immobilized on magnetic nanoparticles modified with aminopropyl triethoxysilane (APTES) using a cross-linking agent, glutaraldehyde. The immobilized enzyme showed more than a three-fold increase in thermal stability and retained 90% activity after the fifth cycle. The authors also found that the immobilized enzyme had a better affinity towards its substrate. The starch–asparagine food model of 2% (*w/v*) L-asparagine and 2% starch (*w/v*) (1:1) was treated with equal dose (18 U) of free or immobilized enzyme for 30 min and then heated at 180 °C for 5 min. The acrylamide formation in starch–asparagine food treated with free enzyme reduced by 60%, whereas no formation of acrylamide treated with immobilized enzyme was observed compared with the control [125].

In another study, Ravi and Gurunathan (2018) used chitosan-immobilized L-asparaginase to reduce acrylamide in fried kochchi kesel. L-asparaginase from *Aspergillus terreus* was covalently immobilized on chitosan by glutaraldehyde. Kochchi kesel banana slices were soaked in 5 U/mL of free or immobilized enzyme suspension at 60 °C for 20 min and then fried at 180 °C for 25 min. The mitigation of acrylamide in fried kochchi kesel with treatment by chitosan-immobilized L-asparaginase decreased by approximately 49% compared to treatment with the free enzyme [126]. Immobilized L-asparaginase is more effective for acrylamide mitigation in food than the free enzyme because of its superior properties.

Immobilized L-asparaginase is utilized not only in batch processing but also in continuous production processes for fluidized food components for acrylamide mitigation. Li et al. (2020) used agarose sphere-immobilized L-asparaginase to reduce acrylamide in a fluid food model system. The immobilized L-asparaginase (Aga-ASNase) was obtained from an amidation reaction between L-asparaginase and food-grade agarose spheres activated with N-hydroxysuccinimide esters. The immobilized enzymes exhibited superior storage stability and reusability, with 93.21% and 72.25% of the initial activity retained after six consecutive cycles and storage for 28 days, respectively. The effluent flowing from the continuous catalytic process under optimal conditions (96 U, 35 °C, 1 mL/min) was heated at 180 °C in an oil bath for 10 min. Compared with the amount of acrylamide in the untreated system, the amount of acrylamide was reduced by almost 89% when the fluids flowed through the packed bed reactor, with an average residence time of 12 min [127].

## 8. Discussion

Although L-asparaginase can effectively reduce the acrylamide content in food such as fried potato products, bakery products, and coffee, some aspects of its application still need to be further discussed. Current research on the application of L-asparaginase in foods to reduce acrylamide cannot fully reveal the correlation between acrylamide mitigation and the biochemical properties of L-asparaginases. Therefore, comprehensive studies are needed to understand the special properties of L-asparaginase required for its food applications, which may differ from the properties required for its medical application. The application of L-asparaginase in food is limited mainly because of properties of the enzyme, such as poor thermal stability and low enzyme activities, and insufficient contact between enzyme and L-asparagine in solid foods during enzyme treatment.

L-asparaginase, applied in some of the lab-scale studies to inhibit acrylamide formation, was derived from recombinant L-asparaginase or native L-asparaginase, which was the aim to find the enzyme with desirable properties. The production of recombinant L-asparaginase is based on its gene sequence from the database. Although it has the advantage of a simplified production process, recombinant L-asparaginase was usually expressed as fusion protein form (L-asparaginase with a hexa-histidine tag on terminus), facilitating its purification by nickel affinity chromatography, which may affect its biochemical properties [12,20]. The production of native L-asparaginase is obtained through the processes such as screening high-yield strains, optimizing fermentation conditions, and purification, which is a traditional method for enzyme production. The advantage of this method is that unknown native L-asparaginases with desirable properties can be explored from microbes.

Most microbial L-asparaginases have glutaminase activity, which may cause clinical side effects regarding the use of L-asparaginase as a therapeutic agent. Glutaminase, catalyzing the hydrolytic deamination of L-glutamine to yield L-glutamate and ammonia, has an indirect role in imparting or enhancing the flavor profile of foods by increasing the L-glutamate content of intermediate food ingredients [128]. Claeys et al. reported that the addition of glutamine had a strong promoting effect on acrylamide formation [129]. Although it is unknown how its glutaminase activity affects food, the influences of the glutaminase activity of L-asparaginase in food applications are worthy of attention.

Factors that have inhibitory effects on the Maillard reaction can prevent the formation of acrylamide, which is formed between reducing sugars such as glucose and L-asparagine through the Maillard reaction. The oxidoreductase may be an oxidase or dehydrogenase capable of reacting with a reducing sugar as a substrate such as glucose and maltose to reduce the acrylamide content in food [110]. Organic acids, such as citric, acetic, and lactic acid, can block the nucleophilic addition of L-asparagine with a carbonyl compound because of the protonation of L-asparagine amino groups at low pH, preventing the formation of acrylamide [130]. Mono- and divalent cations (e.g., Na^+^ and Ca^2+^) were used to efficiently reduce acrylamide formation, which was postulated that these ions could interact with L-asparagine so that the Schiff base formation was prevented [123]. L-asparaginase in food applications combined with factors mentioned above to efficiently reduce acrylamide formation are also worthy of attention.

Furthermore, L-asparaginase from the perspective of its application in food needs further development in the future. Cost-effective L-asparaginases with desirable properties that will be obtained through the use of statistical optimization methods, less laborious fermentative processes, and convenient extraction and purification strategies, will reduce the economic cost of its application in the food industry. It is necessary to explore thermostable L-asparaginases with ideal characteristics from hitherto unprobed niches or gene banks, because they are more suitable for applications in industrial processes where high temperatures are usually required. L-asparaginase could be engineered using an immobilization strategy or a new technique to enhance its properties, such as stability and recyclability, which will promote its application in the food industry.

## 9. Conclusions

L-asparaginase catalyzes the hydrolysis of L-asparagine, and it is a highly significant enzyme in pharmaceutical and food industries. The application of L-asparaginase in the food industry has drawn more attention from researchers and the public in recent years, because it can mitigate carcinogenic acrylamide in food products with a concomitant impact on public health. A variety of microbial l-asparaginases have substantial differences in biochemical properties, which lays a foundation for its application in food. Treating foods such as fried potato products, bakery products, and coffee with microbial L-asparaginase, prior to heat processing, is a simple solution to alleviate acrylamide formation; the advantage of this method is that it does not alter the nutritional value, appearance, or flavor of the final products. Immobilized L-asparaginase possesses unique properties, such as stability and recyclability, which make its potential application feasible in both batch processing and continuous production processes for fluidized food components in the reduction in acrylamide formation in food. However, there is still much work to be conducted to reduce acrylamide in the food industry with the use of L-asparaginase.

## Figures and Tables

**Figure 1 microorganisms-09-01659-f001:**
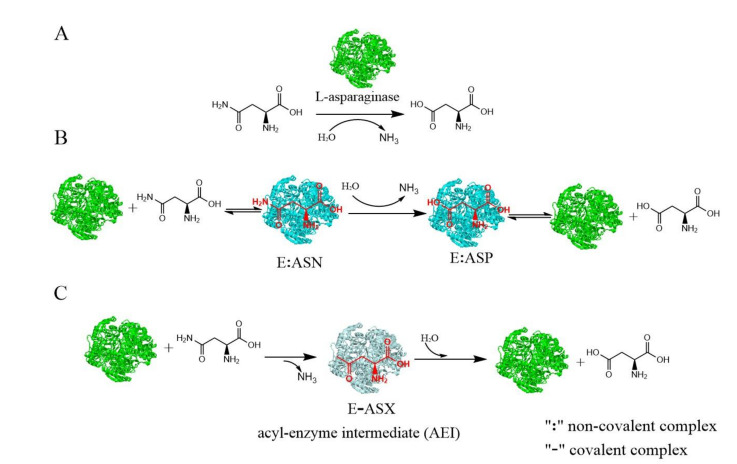
A schematic diagram of the L-asparaginase reaction and its catalytic mechanisms: (**A**) A schematic diagram of the L-asparaginase reaction. (**B**) A schematic diagram of the direct displacement mechanism. (**C**) A schematic diagram of the double-displacement mechanism. Different colors stand for different conformations of the enzyme. (ASN, L-asparagine; ASP, aspartic acid).

**Figure 2 microorganisms-09-01659-f002:**
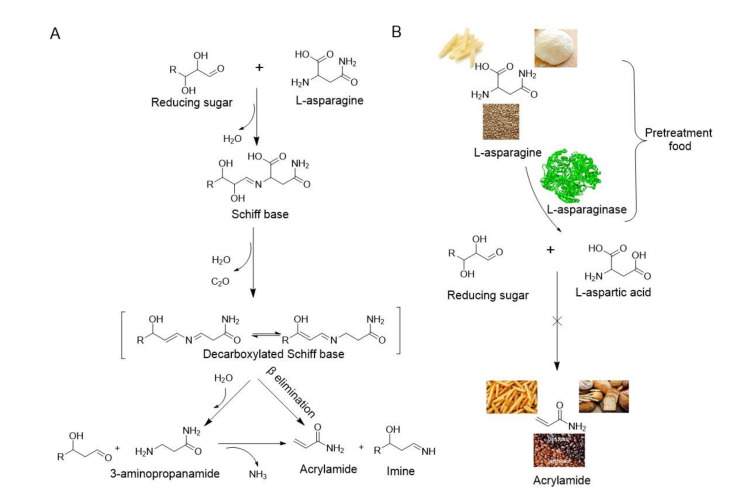
Mechanism of acrylamide formation and acrylamide mitigation through L-asparaginase: (**A**) Acrylamide formed between reducing sugars and L-asparagine; (**B**) Acrylamide mitigation in foods through the conversion of L-asparagine to L-aspartic acid by L-asparaginase.

**Table 1 microorganisms-09-01659-t001:** L-asparaginase from microbes.

Microbe	Strain	Kind of Enzyme	Source	Reference
Archaea	*Thermococcus kodakarensis* KOD1	plant type	cloning gene	[25]
	*Thermococcus kodakarensis* KOD1	type I	cloning gene	[40,41,42]
	*Thermococcus zilligii* AN1 TziAN1_1	–	cloning gene	[12]
	*Pyrobaculum calidifontis*	–	cloning gene	[43]
	*Pyrococcus yayanosii* CH1	–	cloning gene	[44]
Bacteria	*Acinetobacter soli* Y-3	–	cloning gene	[45]
	*Bacillus subtilis* WSA3	extracellular	marine sponges	[46]
	*Bacillus sonorensis*	–	cloning gene	[47]
	*Bacillus licheniformis*	extracellular	soil	[48]
	*Bacillus* sp. SL-1	–	cloning gene	[6]
	*Bacillus brevis*	intracellular	soil	[49]
	*Bacillus subtilis* KDPS1	extracellular	soil	[50]
	*Stenotrophomonas maltophilia* EMCC2297	extracellular	soil	[38]
	*Lactobacillus casei* subsp.casei ATCC 393	–	cloning gene	[51]
	*Pseudomonas otitidis* *Enterobacter cloacae* *Ochrobactrum anthropi* *Escherichia fergusonii*	periplasmic	soils and water	[52]
	*Pseudomonas oryzihabitans*	extracellular	plant	[53]
	*Halomonas elongate* IBRC-M10216	–	cloning gene	[54]
	*Rhizobium etli*	–	cloning gene	[29]
Actinomycetes	*Streptomyces fradiae* NEAE-82	extracellular	soil	[55]
	*Streptomyces rochei* subsp. chromatogenes NEAE-K	extracellular	soil	[35]
	*Streptomyces ansochromogenes* UFPEDA 3420	extracellular	collection	[56,57]
	*Streptomyces* spp.	extracellular	soil	[58]
	*Streptomyces brollosae* NEAE-115	extracellular	soil	[59]
	*Streptomyces labedae* VSM-6	extracellular	marine sediment	[60]
	*Nocardiopsis alba* NIOT-VKMA08	extracellular	marine sediment	[61]
Fungi	*Penicillium crustosum* NMKA 511	extracellular	soil	[19]
	*Penicillium* sp.–encoded 2DSST1	extracellular	soil	[62]
	*Penicillium* sp. T6.2 Fusarium sp. T22.2	extracellular	collection	[63]
	*Penicillium simplicissimum* *Dothiodeomycetes* sp. *Ascomycota* sp. *Fusarium oxysporum*	extracellular	plants	[64]
	*Fusarium equiseti* AHMF4	extracellular	soil	[65]
	*Aspergillus oryzae* CCT 3940	extracellular	collection	[37]
	*Aspergillus fumigatus*	extracellular	plant	[66]
	*Aspergillus oryzae* CCT 3940	extracellular	collection	[67]
	*Sarocladium strictum*	extracellular	soil	[68]
	*Trichosporon asahii* IBBLA1	extracellular	soil and mosses	[69]
Yeast	*Leucosporidium muscorum* CRM 1648	intracellular	marine sediment	[39]
	*Leucosporidium scotti* L120	-	marine sediment	[70]
	*Sarocladium* sp. AG90	extracellular	soil	[71]
	*Yarrowia lipolytica* DSM3286	extracellular	collection	[72]
	*Saccharomyces cerevisiae* BY4741	-	cloning gene	[73]
Algae	*Spirulina maxima*	extracellular	collection	[74,75]
	*Chlorella vulgaris*	intracellular	water and soil	[76]

**Table 2 microorganisms-09-01659-t002:** Purification methods and biochemical characteristic of L-asparaginase produced by microbes.

Strains	Purification				Characterization					References
	Method	SA (IU/mg)	PF	PY (%)	Opt. pH	Opt. temp (°C)	K_m_ (mM)	V_max_/K_cat_(s^−1^) ^1^	Mol. Wt. (kDa) (Structural Form)	
*Thermococcus kodakarensis*	Superdex 200 10/300 GL gel column	767 μmol/min/mg	-	-	7.0	85	3.1	833 μmol/min/mg	70 (homodimer)	[25]
*Pyrobaculum calidifontis*	Superdex 200 10/300 GL gel filtration column	-	-	-	6.5	at or above 100	4.5	355 μmol/min/mg	65 (dimer)	[43]
*Pyrococcus yayanosii* CH1	nickel affinity column	1483.81	50.62	60.92	8.0	95	6.5	2929 μmol/min	72.2 (homodimer)	[44]
*Thermococcus zilligii* AN1 TziAN1_1	nickel affinity column	5278	-	-	8.5	90	6.08	3267	71 (homodimer)	[12]
*Lactobacillus casei* subsp.casei ATCC 393	nickel affinity column	0.419	9.78	99.8	6	40	0.012	1.576 mM/min	35 (monomer)	[51]
*Pseudomonas fluorescens* MTCC 8127	nickel affinity column	26	4.12	85.4	7.5	37	50	4.032 M/min	140 (homotetramer)	[94]
*Lactobacillus reuteri* DSM 20016	nickel affinity column	0.63	10.5	92	6	30	0.3332	14.06 mM/min	35 (monomer)	[95]
*Halomonas elongate* IBRC-M10216	nickel affinity column	1510	14.5	27	6–9	37	5.6	2.2 μmol/min	70–80 (homodimer)	[54]
*Bacillus brevis*	sulphopropyl Sephadex column	9.89	89.9	15	6.5	37	35	0.77 IU	32 (monomer)	[49]
*Bacillus subtilis* sp. KDPS-1	ammonium sulphate precipitation and DEAE column	97.04	12.11	84.89	5.0	37	-	-	97.4 -	[50]
*Pseudomonas otitidis*	ammonium sulphate precipitation, DEAE-cellulose Column and Sephadex G-100 Column	107.84	151.88	38.9	7.5	40	-	-	205 (homohexamer)	[96]
*Aquabacterium* sp. A7-Y (abASNase2)	nickel affinity column	458.9	-	-	9.0	60	1.8	241.9	33 (monomer)	[20]
*Penicillium crustosum* NMKA 511	ammonium sulphate precipitation, DEAE-Sephadex column and Sephadex G-100 column	9.84	6.47	36.3	6.67	36.9	3.79	499.8 µmol/min/mg	41.3 and 44.4 (heterodimer)	[19]
*Aspergillus oryzae* CCT 3940	ammonium sulphate precipitation, Q Sepharose™ column, SP Sepharose™ column and CM Sepharose™ column	282	28.6	6	5–8	40–50	2.10	35.8 U/mL	115 -	[37,67]
*Saccharomyces cerevisiae*	nickel affinity column and PD-10 Desalting column	-	-	-	8.6	40	0.075 ^2^	0.042 μmol/min	41.4 -	[73]
*Candida utilis* ATCC9950	acetone precipitation and Q-sepharose column	7853	10.02	82.7	-	-	-	-	40 -	[97]
*Spirulina maxima*	ammonium sulphate precipitation and Sephadex G-200 column	19.1	10.91	86.45	8.5	37	-	-	- -	[74]

SA, specific activity; PY, protein yield; PF, purification fold; Opt. pH, optimum pH; Opt. temp., optimum temperature; Mol. Wt., molecular weight; K_m_, Michaelis constant. ^1^: V_max_ was replaced with K_cat_ in some references and V_max_ employed different unit in different references. ^2^: L-asparaginase 1 from *Saccharomyces cerevisiae* has allosteric behavior similar to type I enzymes, but with a low K _0.5_ = 75 μM as for therapeutic type II.

## Data Availability

Not applicable.

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
