# Peer review of "Microbial L-asparaginase for Application in Acrylamide Mitigation from Food: Current Research Status and Future Perspectives"

_microorganisms, 2021, doi:10.3390/microorganisms9081659_

Round 1

Reviewer 1 Report

The authors provide a comprehensive review regarding the status of application of asparaginase to mitigate acrylamide levels in foods.

There are some point that would require further discussion, specifically regarding the current limitations of asparaginase application:

All mentioned studies are laboratory-scale or pilot studies, but a real life application in industry appears to be lacking. Did you also search the patent literature of there may be industry-scale applications?

The costs of asparaginase treatment are not transparently considered. In the conclusion, it is mentioned that it might be cost-effective, but this is not found in the data of the paper. I actually rather doubt that asparaginase treatment could be more widely applied in coffee industry, with coffee being the largest food commodity worldwide. This would require huge amounts of asparaginase, and regarding the extremely low price of coffee would probably not be feasible, apart from logistic problems in the rural areas of the coffee producing countries. Coffee is not so intesting for mitigation anyway, because the acrylamide contents are typically below the EU limits.

Another point that would need appreciation is the sensory changes of the foods by the treatment? Is there any data on the taste of the products?

Finally, there are other pathways for acrylamide formation that are not influenced by asparaginase. Perhaps this is the reason, why 100% mitigation was never possible.

Some further specific and editorial remarks:

Line 16: of should read between

Line 17: high should read higher

Figure 1: enlarge font size, very difficult to read

Figure 1: explain abbreviations in legend (ASN etc)

Line 187: “these optimized conditions”: which ones?

Line 202 and throughout including tables: please italicize all binominal nomenclature names of microorganisms

Line 281: is this µg/kg bodyweight?

Line 384: “canefora robusta” should read “canephora var. robusta”.

Line 390: mainly should read exclusively (green coffee is free of acrylamide)

Line 404: is it roasted or green coffee?

Section 7: some regulatory aspects could be added. Is the asparaginase treatment allowed by food laws?

Line 473: neither the commercial significance nor the economical viability were demonstrated in the article. Please back up these claims or delete.

Author Response

Dear Editors and Reviewers:

Thank you for your letter and for the reviewers’ comments concerning our manuscript entitled “Microbial L-asparaginase for application in acrylamide mitigation from food: Current research status and future perspectives” (ID: microorganisms-1312154). Those comments are all valuable and very helpful for revising and improving our paper. We have studied comments carefully and have made correction which we hope to meet with approval. It marked revised portion in red in the paper. Point by point responses to the reviewers’ comments are listed below.

Best regards,

Ruiying Jia, Xiao Wan, Xu Geng*, Deming Xue, Zhenxing Xie and Chaoran Chen**

* Correspondence: gengxu@henu.edu.cn

** Correspondence: kfccr@126.com

Reviewer

Comments and Suggestions for Authors

  1. The authors provide a comprehensive review regarding the status of application of asparaginase to mitigate acrylamide levels in foods.

Response: Thank you very much for your valuable comments.

  1. There are some point that would require further discussion, specifically regarding the current limitations of asparaginase application:

Response: Thank you for your valuable comments. Based on your comments, we added the revised content on page 18 line 521-536. The details are as follows: Although L-asparaginase can effectively reduce the acrylamide content in food such as fried potato products, bakery products, and coffee, there are still some problems that need to be resolved. Current research on the application of L-asparaginase in foods to reduce acrylamide cannot fully reveal the correlation between acrylamide mitigation and the biochemical properties of L-asparaginases. Therefore, comprehensive studies are needed to understand the special properties of L-asparaginase required for its food applications, which may differ from the properties required for its medical application. Most microbial L-asparaginases have glutaminase activity, which may cause clinical side effects regarding the use of L-asparaginase as a therapeutic agent. Although the influences of the glutaminase activity of L-asparaginase in food applications are worthy of attention, it is unknown how its glutaminase activity affects food. The application of L-asparaginase in food is limited mainly because of properties of the enzyme, such as poor thermal stability, and insufficient contact between enzyme and L-asparagine in solid foods during enzyme treatment. The research mentioned in this review primarily comprise laboratory-scale studies which are the basis to improve the application of L-asparaginase in food industry processes.

  1. All mentioned studies are laboratory-scale or pilot studies, but a real life application in industry appears to be lacking. Did you also search the patent literature of there may be industry-scale applications?

Response:  Thank you for your valuable comments. All studies mentioned in this review all are laboratory-scale. Based on your suggestion, we added the revised content on page 16 line 441-446. The details are as follows: Furthermore, the application of L-asparaginase to reduce acrylamide in food has been patented. The patent invented by Zyzak et al. indicates that adding L-asparaginase to food material can reduce the level of asparagine and acrylamide in the food product [129]. Another patent showed that, compared to food without L-asparaginase, the addition of an amount of L-asparaginase in food material before heating is effective in reducing the level of acrylamide in the final product [130].

  1. The costs of asparaginase treatment are not transparently considered. In the conclusion, it is mentioned that it might be cost-effective, but this is not found in the data of the paper. I actually rather doubt that asparaginase treatment could be more widely applied in coffee industry, with coffee being the largest food commodity worldwide. This would require huge amounts of asparaginase, and regarding the extremely low price of coffee would probably not be feasible, apart from logistic problems in the rural areas of the coffee producing countries. Coffee is not so intesting for mitigation anyway, because the acrylamide contents are typically below the EU limits.

Response: Thank you for your valuable comments. We agree with you. In the future, cost-effective L-asparaginase is the basis of its wide application in food. However, the solution to this problem needs furthermore work by researcher.

  1. Another point that would need appreciation is the sensory changes of the foods by the treatment? Is there any data on the taste of the products?

Response: Thank you for your valuable comments. We checked the references again. There are few reports about data on the taste of the products in literature. According to the references, L-asparaginase treatment is a simple and effective way to reduce acrylamide in food without affecting sensory or nutritional properties of final products.

  1. Finally, there are other pathways for acrylamide formation that are not influenced by asparaginase. Perhaps this is the reason, why 100% mitigation was never possible.

Response: Thank you for your valuable comments. We believe that the reason that acrylamide content in food can’t be completely eliminated may be that there are other pathways to form acrylamide formation without influencing by L-asparaginase as you mentioned or that L-asparaginase can’t completely hydrolyze L-asparagine in solid food during the treatment. About this point, we add content to the manuscript on page 18, line 521-536.

  1. Some further specific and editorial remarks:

Line 16: of should read between

Line 17: high should read higher

Figure 1: enlarge font size, very difficult to read

Figure 1: explain abbreviations in legend (ASN etc)

Line 187: “these optimized conditions”: which ones?

Line 202 and throughout including tables: please italicize all binominal nomenclature names of microorganisms

Line 281: is this µg/kg bodyweight?

Line 384: “canefora robusta” should read “canephora var. robusta”.

Line 390: mainly should read exclusively (green coffee is free of acrylamide)

Line 404: is it roasted or green coffee?

Response: Thank you for your valuable comments. According to your suggestions, we have revised all the corresponding parts of our manuscript.

  1. Section 7: some regulatory aspects could be added. Is the asparaginase treatment allowed by food laws?

Response: Thank you for your valuable comments. Based on your suggestion, we added the revised content on page 3 line 78-81. The details are as follows: Acrylaway® and PreventAse® are commercially available forms of L-asparaginase from fungi (Aspergillus oryzae and Aspergillus niger) [5,22], which are safe and are used as food additives by FAO/WHO Expert committee [22,23]. L-Asparaginase has been granted “generally recognized as safe” (GRAS) status from the FDA [16].

But the research of immobilized enzyme is mainly laboratory scale research at present, so the legal regulation of immobilized enzyme is still in the development.

  1. Line 473: neither the commercial significance nor the economical viability were demonstrated in the article. Please back up these claims or delete.

Response: Thank you for your valuable comments. We deleted this part (which is of commercial significance and economically viable) in the manuscript.

Reviewer 2 Report

The review deals with the use of asparaginase in food manufacturing, aiming to lower acrylamide formation during cooking.

The topic is very interesting, but the text is overall poorly exhaustive, lacks some information, the style and organization (a more fluent presentation and organization of the reported information), and the English language need extensive improvements.

Same flaws among others: authors omitted to describe the well-known glutaminase activity of most bacterial asparaginases (whose effects on food might deserve attention, as well). Moreover, when reporting the combined enzymatic and bleaching treatment, they do not explain either what the latter consists, or its possible impact in terms of food chemical modification.

Author Response

Dear Editors and Reviewers:

Thank you for your letter and for the reviewers’ comments concerning our manuscript entitled “Microbial L-asparaginase for application in acrylamide mitigation from food: Current research status and future perspectives” (ID: microorganisms-1312154). Those comments are all valuable and very helpful for revising and improving our paper. We have studied comments carefully and have made correction which we hope to meet with approval. It marked revised portion in red in the paper. Point by point responses to the reviewers’ comments are listed below.

Best regards,

Ruiying Jia, Xiao Wan, Xu Geng*, Deming Xue, Zhenxing Xie and Chaoran Chen**

* Correspondence: gengxu@henu.edu.cn

** Correspondence: kfccr@126.com

Reviewer

Comments and Suggestions for Authors

  1. The review deals with the use of asparaginase in food manufacturing, aiming to lower acrylamide formation during cooking.

Response: Thank you very much for your valuable comments.

  1. The topic is very interesting, but the text is overall poorly exhaustive, lacks some information, the style and organization (a more fluent presentation and organization of the reported information), and the English language need extensive improvements.

Response: Thank you for your valuable comments. In order to enrich the content of the paper, we added the contents to the article on page 6-7, page 16 and page 18 of the manuscript. At the same time, we have used English Editing Services of MDPI to improve the language of our manuscript, so as to make the manuscript more fluent and organized.

  1. Same flaws among others: authors omitted to describe the well-known glutaminase activity of most bacterial asparaginases (whose effects on food might deserve attention, as well). Moreover, when reporting the combined enzymatic and bleaching treatment, they do not explain either what the latter consists, or its possible impact in terms of food chemical modification.

Response: Thank you for your valuable comments. Based on your comments, we added the revised content on page 18 line 521-536. The details are as follows: Most microbial L-asparaginases have glutaminase activity, which may cause clinical side effects regarding the use of L-asparaginase as a therapeutic agent. Although the influences of the glutaminase activity of L-asparaginase in food applications are worthy of attention, it is unknown how its glutaminase activity affects food.

 Regarding bleaching, we are sorry that the word (bleaching) is a spelling error. We have replaced bleaching with blanching in the manuscript.

Reviewer 3 Report

The authors reviewed the application of microbial L-asparaginase to mitigate acrylamide in food. There are concerns about the novelty of this manuscript, since similar review articles have already been published:

Munir, N., Zia, M. A., Sharif, S., Tahir, I. M., Jahangeer, M., Javed, I., ... & Shah, S. M. A. (2019). L-Asparaginase potential in acrylamide mitigation from foodstuff: a mini-review. Prog. Nutr, 21(3), 498-506.

Jha, S. K., Pasrija, D., Sinha, R. K., Singh, H. R., Nigam, V. K., & Vidyarthi, A. S. (2012). Microbial L-asparaginase: a review on current scenario and future prospects. International journal of pharmaceutical sciences and research, 3(9), 3076.

Author Response

Dear Editors and Reviewers:

Thank you for your letter and for the reviewers’ comments concerning our manuscript entitled “Microbial L-asparaginase for application in acrylamide mitigation from food: Current research status and future perspectives” (ID: microorganisms-1312154). Those comments are all valuable and very helpful for revising and improving our paper. We have studied comments carefully and have made correction which we hope to meet with approval. It marked revised portion in red in the paper. Point by point responses to the reviewers’ comments are listed below.

Best regards,

Ruiying Jia, Xiao Wan, Xu Geng*, Deming Xue, Zhenxing Xie and Chaoran Chen**

* Correspondence: gengxu@henu.edu.cn

** Correspondence: kfccr@126.com

Reviewer

Comments and Suggestions for Authors

  1. The authors reviewed the application of microbial L-asparaginase to mitigate acrylamide in food. There are concerns about the novelty of this manuscript, since similar review articles have already been published:

Munir, N., Zia, M. A., Sharif, S., Tahir, I. M., Jahangeer, M., Javed, I., ... & Shah, S. M. A. (2019). L-Asparaginase potential in acrylamide mitigation from foodstuff: a mini-review. Prog. Nutr, 21(3), 498-506.

Jha, S. K., Pasrija, D., Sinha, R. K., Singh, H. R., Nigam, V. K., & Vidyarthi, A. S. (2012). Microbial L-asparaginase: a review on current scenario and future prospects. International journal of pharmaceutical sciences and research, 3(9), 3076.

Response: Thank you for your valuable comments. When we were writing this manuscript, we have noticed there were several of related reviews has been published, as you mentioned. The application of microbial L-asparaginase to mitigate acrylamide has become a research hotspot in the field of food safety. In recent years, a large number of research articles have been published in this field, and some new related research progress has appeared. According to the recently published literature, we analyzed and summarized the new progress of the application of microbial L-asparaginase to mitigate acrylamide, and described it in detail; and we also put forward our own views about it in the manuscript. Therefore, we think this manuscript has some novelty. Thank you again for your suggestion.

Round 2

Reviewer 2 Report

The revised manuscript has been improved, but my concerns about the organization remain overall unchanged. There are many unrelated and/or poorly pertinent information (the description of production and purification are too long) and the paragraph dealing with the true topic is relatively short and authors should evaluate to change the style, avoiding reporting as a list (without comments or contextualization) the state of the art.

L442-444 and L445-447: what's the difference between the two sentences? The concept seems to be exactly the same.

L529-533: that sentence about the glutaminase activity of ASN looks out of place in the conclusions section. Instead, this information should be reported elsewhere (for example describing the activities and the biochemical properties of the enzyme). Moreover, it is true that the effects of such a secondary catalytic activity have not been assessed on food, but the effects of glutaminase on food are well known. This provides arguments to be discussed, even as hypotheses. (see 10.1002/fsn3.1426) The authors seem to have done no effort in this direction, to improve the manuscript.

Overall, the conclusions section should be concise and conclusive (and include perspectives), not a long list of point-sentences sounding like a summary of the text.

Author Response

Dear Editors and Reviewers:

Thank you for your letter and for the reviewers’ comments concerning our manuscript entitled “Microbial L-asparaginase for application in acrylamide mitigation from food: Current research status and future perspectives” (ID: microorganisms-1312154). Those comments are all valuable and very helpful for revising and improving our paper. We have studied comments carefully and have made correction which we hope to meet with approval. It marked revised portion in red in the paper. Point by point responses to the reviewers’ comments are listed below.

Best regards,

Ruiying Jia, Xiao Wan, Xu Geng*, Deming Xue, Zhenxing Xie and Chaoran Chen**

* Correspondence: gengxu@henu.edu.cn

** Correspondence: kfccr@126.com

Reviewer

Comments and Suggestions for Authors

  1. The revised manuscript has been improved, but my concerns about the organization remain overall unchanged. There are many unrelated and/or poorly pertinent information (the description of production and purification are too long) and the paragraph dealing with the true topic is relatively short and authors should evaluate to change the style, avoiding reporting as a list (without comments or contextualization) the state of the art.

Response: Thank you for your valuable comments. According to the correlation between the content and the topic of the manuscript, we adjusted the organizational structure of the manuscript. We deleted the parts of Section 4 (Production of L-asparaginase) and Section 5 (Purification and biochemical properties of microbial L-asparaginase) and added parts of Section 6 (Application of microbial L-asparaginase in food; page 11 line 309-316, page 12 line 358-363, page 13 line 407-418, and page13-14 line 419-456) and created Section 8 (8. Discussion; page 14-15 line 502-542) of the manuscript. At the same time, we merged Tables 1 to 5 into a new Table 1, deleted Tables 6 and 7, and changed the original Table 8 into Table 2 of the manuscript.

  1. L442-444 and L445-447: what's the difference between the two sentences? The concept seems to be exactly the same.

Response: Thank you for your valuable comments. You were right. We deleted the entire part from the manuscript due to its concept being the same as other contents.

  1. L529-533: that sentence about the glutaminase activity of ASN looks out of place in the conclusions section. Instead, this information should be reported elsewhere (for example describing the activities and the biochemical properties of the enzyme). Moreover, it is true that the effects of such a secondary catalytic activity have not been assessed on food, but the effects of glutaminase on food are well known. This provides arguments to be discussed, even as hypotheses. (see 10.1002/fsn3.1426) The authors seem to have done no effort in this direction, to improve the manuscript.

Response: Thank you for your valuable comments. We have read the article you mentioned and got the new ideas from it. We discussed this problem and added the corresponding content in Section 8 (Discussion; page 15 line: 513-520) the manuscript.

  1. Overall, the conclusions section should be concise and conclusive (and include perspectives), not a long list of point-sentences sounding like a summary of the text.

Response: Thank you for your valuable comments. We changed the conclusion of the manuscript based on the concise and conclusive principle (page 15-16 line 544-557). The details are as follows: L-asparaginase catalyzes the hydrolysis of L-asparagine, and it is a highly significant enzyme in pharmaceutical and food industries. The application of L-asparaginase in the food industry has drawn more attention from researchers and the public in recent years, as it can mitigate carcinogenic acrylamide in food products with a concomitant impact on public health. A variety of microbial L-asparaginases have substantial differences in biochemical properties, which lays a foundation for its application in food. Treating foods such as fried potato products, bakery products, and coffee with microbial L-asparaginase, prior to heat processing, is a simple solution to alleviating acrylamide formation; the advantage of this method is that it does not alter the nutritional value, appearance, or flavor of final products. Immobilized L-asparaginase possesses unique properties, such as stability and recyclability, which make its potential application both in batch processing and in continuous production processes for fluidized food components in the reduction of acrylamide formation in food. However, there is still much work to be conducted to reduce acrylamide in the food industry with the use of L-asparaginase.

Reviewer 3 Report

N.A.

Author Response

Dear Editors and Reviewers:

Thank you for your letter and for the reviewers’ comments concerning our manuscript entitled “Microbial L-asparaginase for application in acrylamide mitigation from food: Current research status and future perspectives” (ID: microorganisms-1312154). Those comments are all valuable and very helpful for revising and improving our paper. We have studied comments carefully and have made correction which we hope to meet with approval. It marked revised portion in red in the paper. Point by point responses to the reviewers’ comments are listed below.

Best regards,

Ruiying Jia, Xiao Wan, Xu Geng*, Deming Xue, Zhenxing Xie and Chaoran Chen**

* Correspondence: gengxu@henu.edu.cn

** Correspondence: kfccr@126.com

Reviewer

Comments and Suggestions for Authors

  1. The authors reviewed the application of microbial L-asparaginase to mitigate acrylamide in food. There are concerns about the novelty of this manuscript, since similar review articles have already been published:

Munir, N., Zia, M. A., Sharif, S., Tahir, I. M., Jahangeer, M., Javed, I., ... & Shah, S. M. A. (2019). L-Asparaginase potential in acrylamide mitigation from foodstuff: a mini-review. Prog. Nutr, 21(3), 498-506.

Jha, S. K., Pasrija, D., Sinha, R. K., Singh, H. R., Nigam, V. K., & Vidyarthi, A. S. (2012). Microbial L-asparaginase: a review on current scenario and future prospects. International journal of pharmaceutical sciences and research, 3(9), 3076.

Response: Thank you for your valuable comments. We have carefully read the two articles you mentioned. We thought over your comments and believed that it was high valuable to our manuscript. We added several of the revised content on the manuscript following except the original Section 7 (7. application of immobilized L-asparaginase in food): 1) adding the content on the application of L-asparaginase in food from the patent literature (page 11 line 309-316, page 12 line 358-363, and page 13 line 407-418); 2) adding the content on the application of L-asparaginase in food industry process (6.4. Industrial processes, page13-14 line 419-456); 3) putting forward some ideas about the application of L-asparaginase in food in Section 8 (8. Discussion; page 14-15 line 502-542). Thank you again for your comments.
